# Haze Occurrence Caused by High Gas-to-Particle Conversion in Moisture Air under Low Pollutant Emission in a Megacity of China

**DOI:** 10.3390/ijerph19116405

**Published:** 2022-05-25

**Authors:** Qingxia Ma, Weisi Wang, Dexin Liu, Rongke Zhao, Jingqi Zhao, Wanlong Li, Yanfang Pan, Daizhou Zhang

**Affiliations:** 1Key Laboratory of Geospatial Technology for the Middle and Lower Yellow River Regions, Ministry of Education, College of Geography and Environmental Science, Henan University, Kaifeng 475004, China; mqx@henu.edu.cn (Q.M.); dxliu@vip.henu.edu.cn (D.L.); zhaojingqi@henu.edu.cn (J.Z.); liwanlong@henu.edu.cn (W.L.); 2Henan Key Laboratory of Integrated Air Pollution Control and Ecological Security, Kaifeng 475004, China; 3Henan Ecological and Environmental Monitoring Center, Zhengzhou 450007, China; cameraart@126.com; 4Henan Kaifeng College of Science Technology and Communication, Kaifeng 475004, China; zrk_heart@163.com; 5Faculty of Environmental and Symbiotic Sciences, Prefectural University of Kumamoto, Kumamoto 862-8502, Japan

**Keywords:** haze, gas-to-particle conversion, high RH, urban pollution, emission restriction

## Abstract

Haze occurred in Zhengzhou, a megacity in the northern China, with the PM_2.5_ as high as 254 μg m^−3^ on 25 December 2019, despite the emergency response measure of restriction on the emission of anthropogenic pollutants which was implemented on December 19 for suppressing local air pollution. Air pollutant concentrations, chemical compositions, and the origins of particulate matter with aerodynamic diameter smaller than 2.5 µm (PM_2.5_) between 5–26 December were investigated to explore the reasons for the haze occurrence. Results show that the haze was caused by efficient SO_2_-to-suflate and NO_x_-to-nitrate conversions under high relative humidity (RH) condition. In comparison with the period before the restriction (5–18 December) when the PM_2.5_ was low, the concentration of PM_2.5_ during the haze (19–26 December) was 173 µg m^−3^ on average with 51% contributed by sulfate (31 µg m^−3^) and nitrate (57 µg m^−3^). The conversions of SO_2_-to-sulfate and NO_x_-to-nitrate efficiently produced sulfate and nitrate although the concentration of the two precursor gases SO_2_ and NO_x_ was low. The high RH, which was more than 70% and the consequence of artificial water-vapor spreading in the urban air for reducing air pollutants, was the key factor causing the conversion rates to be enlarged in the constriction period. In addition, the last 48 h movement of the air parcels on 19–26 December was stagnant, and the air mass was from surrounding areas within 200 km, indicating weather conditions favoring the accumulation of locally-originated pollutants. Although emergency response measures were implemented, high gas-to-particle conversions in stagnant and moisture circumstances can still cause severe haze in urban air. Since the artificial water-vapor spreading in the urban air was one of the reasons for the high RH, it is likely that the spreading had unexpected side effects in some certain circumstances and needs to be taken into consideration in future studies.

## 1. Introduction

Severe haze with high PM_2.5_ concentration frequently occurs in the North China Plain (NCP) of China, imposing serious burden to the environment and threats to public health [1,2,3]. In order to suppress the heavy air pollution, the Chinese government has taken a series of control measures, including cutting down pollutant emissions, limiting car-use, constructing clean-coal power plants, prohibiting open burning of crop residues during harvest seasons, etc. [4,5]. As a result, the annual mean PM_2.5_ concentration significantly reduced nationwide in the past several years [6]. Yet, PM_2.5_ concentration in many megacities of the NCP became high sometimes [7].

Previous studies have shown that the formation of secondary inorganic aerosols (SIAs, the sum of sulfate (SO_4_^2−^), nitrate (NO_3_^−^), and ammonium (NH_4_^+^)) was one of the major reasons causing the haze pollution with its 40–70% contribution to PM_2.5_ in the NCP on an average basis [8,9]. The nonlinear relation between the decreases of sulfate and nitrate and their precursors has not been well understood, although the complex formation and atmospheric transformation processes are likely mediated by meteorological conditions [10,11]. For example, high relative humidity (RH > 60%) facilitates SIA formation via heterogeneous reactions and aqueous-phase reactions [12,13]. O_3_ oxidation is a pathway for nitrate and sulfate formation under alkaline conditions [14,15]. Moisture conditions favored the formation of SIAs via heterogeneous reactions, and dry conditions inhibited gas-to-particle partitioning [8,16,17].

The current pollution control in China enabled substantial reduction of air pollutants from transportation, industrial, and human activities. However, haze still occurred sometimes in winter, indicating that the reduction did not stop the formation of haze [18,19,20]. The nonlinear relationships between the reductions of particulate matter (PM) and their precursors, unfavorable meteorological conditions, and enhanced secondary production were reported as the causes for the haze [10,21,22,23,24,25].

Zhengzhou is a megacity, with a population of approximately 12.6 million and land area of about 7567 km^2^, which has high coal consumption, large transportation systems, and concentrative industries [26,27]. The city, located in the southern part of the NCP, is surrounded by other densely populated and industrialized cities. Serious air pollution and low visibility frequently occurred in the city (Figure 1, [22,28,29,30]). Due to the variety of energy structures and economy levels from city to city in China, one-city-one air pollution control-policy is urgently required [10,23].

Zhengzhou implemented emergency response measures, such as shutting down and restricting industrial activities, prohibiting open burning, reducing heavy vehicles on roads, and spreading water vapor on public roads since 19 December 2019 to suppress the occurrence of haze [31]. However, a severe haze occurred between 19–26 December. The purpose of this study is to explore the reasons for the haze occurrence with the routine data from an urban monitoring site on air pollutant concentration, chemical composition, and the modelled origins of the PM_2.5_ under control measure conditions and non-control measures conditions.

## 2. Materials and Methods

### 2.1. Observation Site and Instruments

The concentrations of air pollutants including PM_2.5_, PM_10_, O_3_, SO_2_, NO_x_, and CO and meteorological conditions including pressure, temperature, RH, wind speed, wind direction, and precipitation are routinely measured at the environment monitoring supersite of Henan province in Zhengzhou (34.76° N, 113.70° E; Figure 1). The instruments are a series of pollutant detectors (5030, 5014i, 49I, 42I, 43I, 48I; Thermo Fisher Scientific, Waltham, MA, USA) and a weather station (Vantage VUE, Davis Instruments, Inc., Hayward, CA, USA) at a time resolution of 5 min. One-hour average of the air pollutants and weather conditions was applied.

The hourly mass concentrations of major water-soluble inorganic ions, including NH_4_^+^, Na^+^, K^+^, Ca^2+^, Mg^2+^, SO_4_^2−^, NO_3_^−^, and Cl^−^ in aerosols, were measured by an online analyzer for Monitoring of AeRosols and GAses (MARGA, model ADI 2080 Applikon Analytical B. V. Corp., Delft, The Netherlands) at a flow rate of 16.7 L min**^−^**^1^ [32,33]. A thermo-optical OC/EC analyzer (Model RT-4, Sunset Lab., Los Angeles, CA, USA) measured organic carbon (OC) and elemental carbon (EC) in aerosol particles. Elements in PM_2.5_, including K, Ca, Pb, Se, Cr, Zn, Cu Ni, Fe, Mn, Ti, V, Ba, As, Co, Mo, Sc, Br, Si, and Al, were quantified by an online analyzer for monitoring (AMMS-100, Focused Photonics Inc., Hangzhou, China). The formulas of major components in the PM_2.5_ and the conversion rates of SO_2_ to sulfate (SOR) and NO**_x_** to nitrate (NOR) are listed in Table 1.

In this study, the observational data between 5–26 December of 2019 were utilized. Three haze events in the observation period occurred during 5–9 December (PP1), during 11–17 December (PP2) before restriction, and during December 19 and 26 with restriction.

### 2.2. PMF Model

The origins of PM_2.5_ were analyzed with the positive matrix factorization (PMF) model of USEPA version 5.0. The model is a widely used receptor model, has high efficiency and convenience without the use of pollution discharge conditions, and allows additional constraints to be added into the factor profiles or factor contributions to reduce result uncertainties [42,43]. Input factors for the model included eight water soluble ions (e.g., Na^+^, NH_4_^+^, K^+^, Mg^2+^, Ca^2+^, NO_3_^−^, SO_4_^2−^, and Cl^−^), and nine target metal elements (e.g., Al, Fe, Ti, Mn, Cu, Zn, Sb, Pb, and Cr) as well as the PM_2.5_ mass concentration. In this study, five source types were tested in the analysis.

### 2.3. Air Mass Backward Trajectories Cluster Analysis

Backward trajectories of air mass arriving at the sample site (34.76° N, 113.70° E) were calculated for 48 h using the off-line HYSPLIT 4 model. Archived meteorological data with a 1° × 1° latitude–longitude grid and 3 h time interval were used for the trajectory calculations and were provided by the US National Centers for Environmental Prediction Global Data Assimilation System (GDAS) (ftp://arlftp.arlhq.noaa.gov/pub/archives/gdas1, accessed on 5 December 2021). The arrival altitude was set at 500 m above ground level (a.g.l.). The model was run each hour during the whole period focused on in this study.

## 3. Results and Discussion

### 3.1. Weather Conditions and Haze Pollution

An extremely severe haze pollution episode occurred between December 19 and 26 after the emergency response measures for heavily polluting weather were implemented on 19 December. The wind speed was usually less than 2 m s^−^^1^ and the RH was mostly greater than 40% between December 19 and 26, indicating that air movement was stagnant. Under these weather conditions, pollutants were hardly diffused. PM_10_ and PM_2.5_ concentrations started to increase from 43 and 28 µg m*^−^*^3^ sharply in the afternoon (13:00–14:00 local time) on 18 December 2019, and the high levels lasted until the morning (10:00–11:00) on December 26 (Figure 2). The maximal mass concentration of PM_10_ and PM_2.5_ reached 332 and 254 μg m*^−^*^3^ at night on 25 December, respectively. The wind speed was 1.4 m s^−1^ and the RH was 89%. SIA concentration was 185 µg m*^−^*^3^, accounting for 73% of PM_2.5_ concentration. Under the stable weather conditions, high SIA formation was the major reason for the extremely high aerosol load. The average mass concentrations of PM_10_ and PM_2.5_ during the haze period were 216.86 and 172.63 μg m*^−^*^3^, indicating the extremely high levels of aerosol pollution. The mass concentration ratios of PM_2.5_/PM_10_ during 19–26 December, i.e., the haze occurrence period (hereafter called HOP) was 0.80, indicating the dominance of PM_2.5_ in the total aerosol mass.

The average concentrations of SO_2_, O_3_, CO, NO, and NO_2_ were 10, 12, 2, 23, and 64 μg m^−3^, respectively, in the HOP. The SO_2_ and NO_2_ concentrations were lower than those during a haze episode between January 12 and 23, 2018 in Zhengzhou, but the O_3_ concentration was similar [16]. The mass concentration of CO also rose with PM_2.5_. The mass concentrations of SO_2_ and O_3_ were high in the daytime, with the maximum concentration in the noon or early afternoon, and low in the nighttime on 18–22 December (Figure 3). The trend in SOR followed that of RH quite well; it was high during the night but low during the day and more SO_2_ was converted during nighttime [10,11,18]. In comparison, the conversion during daytime was small, and SO_2_ concentration was relatively high during the daytime. As O_3_ was formed through photochemical reaction, O_3_ was high in the daytime [12,14,15]. After a rapid increase of NO_2_ concentration in the late afternoon on December 18, NO_2_ concentration stayed high in the nighttime and became low in the late afternoon during 18–22 December. It was attributed to NO_2_ that was oxidized by hydroxyl radical (OH) and high O_3_ to form HNO_3_ in the daytime [13,14,16]. This phenomenon was found by other researchers [14,44]. The concentrations of SO_2_, O_3_, and NO_2_ were approximately stable during other periods. SO_2_ increased ~6%, NO_2_ 43%, CO 107.45%, and NO 220.58% from the time before the haze (i.e., before 18 December) to the time of the HOP.

The increases of different gas pollutants varied with haze development. This phenomenon could be attributed to the differences in the relevant chemical reactions and the responses to emergency response measures. Here we use the concentration of CO as the reference value to investigate the roles of chemical reactions in the variation because CO was produced only from primary emissions and was a very inertial species to the chemical reactions [40,41]. The variability of CO was dominated by emission intensity and atmospheric physical processes (dilution/mix effect). The normalized concentration of PM_2.5_ and precursors by CO can counteract the effect of atmospheric physical processes and represent the contribution of chemical reactions. 

With the development of the haze pollution, the ratio of PM_2.5_/CO rapidly increased between the afternoon of 18 December and 22 December, then slightly increased between 23 December and the morning of 26 December, and finally largely reduced (Appendix A), reflecting an elevated production rate of secondary species in the HOP. The SO_2_/CO and NO_2_/CO ratios decreased with haze pollution development, which was consistent with the substantially increasing PM_2.5_/CO between 19–26 December (Appendix A). This result implies that the air pollution on 19–26 December likely had not been eliminated, although emergency measures had been enacted. A recent study found that, despite emission reductions of 90% across all sectors over Beijing and surrounding provinces, heavily polluted days with daily mean PM_2.5_ higher than 225 µg m^−3^ may not have been eliminated enough to meet national air quality standards [4]. 

### 3.2. Chemical Composition of PM_2.5_ during the HOP

In the HOP, the average concentration of SIAs (including NO_3_^−^, NH_4_^+^, and SO_4_^2−^), OM, EC, minerals, sea salt, and K salt were 102.8, 20.3, 4.3, 6.1, 2.2, and 1.1 μg m^−3^, respectively. SIAs were the major components in the PM_2.5_ and contributed approximately 68%, followed by OM (13%), minerals (4%), EC (3%), sea salt (1%), and K salt (1%) (Figure 2c). Although the mass concentrations of chemical species in PM_2.5_ obviously increased during the HOP, the relative contributions of chemical species to PM_2.5_ varied differently. The contribution of SO_4_^2−^ increased by 4.5%, and that of NH_4_^+^ by 1.0%, whereas the contribution of OM, NO_3_^−^, sea salt, and mineral cations reduced by 8.95%, 3.94%, 3.88%, and 3.20%. The contribution of NO_3_^–^ and SO_4_^2−^ to PM_2.5_ was higher (51%) compared to those in previous studies on PM_2.5_ in Zhengzhou [23,28,45].

RH, T, and O_3_ are key factors regulating the oxidation pathways for nitrate and sulfate formation [46,47,48]. Although SO_2_ and NO_2_ did not directly result in haze occurrence, approximately 119 µg m^−3^ of the 173 µg m^−3^ PM_2.5_ was SIAs, which was related to the two precursor gases during the HOP. 

To identify the factors associated with the efficient formation of the SIAs, we compared the formation of NO_3_^−^ and SO_4_^2−^ under different levels of their precursor gases, RH, and oxidation ratios. NO_3_^−^ and SO_4_^2−^ positively correlated with RH during the HOP (Figure 4a,b). They rapidly increased from 30 μg m^−3^ to 90 μg m^−3^ and from 15 μg m^−3^ to 70 μg m^−3^, respectively, as the RH increased from 70% to 100%, suggesting NO_3_^−^ and SO_4_^2−^ were efficiently produced in moisture air. Regarding the formation of nitrate and sulfate being usually dominated by heterogeneous reactions under high RH (RH > 60%) conditions and by gas-phase reactions at low RH (RH < 30%) conditions [49,50], the efficient production of the two salts in the present case could be attributed to heterogeneous reactions under high RH conditions. High RH favored NO_3_^–^ and SO_4_^2–^ formation and subsequently led to the rapid increase of PM_2.5_. 

It was found that higher SOR led to a rapid increase of SO_4_^2−^ at a given RH, whereas NOR did not. SO_4_^2−^ formation was more closely associated with the secondary conversion rate than NO_3_^−^ formation. Rapid increases in NO_3_^–^ were accompanied by large NOR even at low levels of NOx or by low NOR with high NO_x_ levels (Appendix A). NO_3_^−^ at high NOx levels was produced as efficiently as that of high NOR with low NO_x_ levels. This result highlights that the NO_3_^–^ production was significantly influenced by both NO_x_ concentration and the NOR.

As shown in Figure 4c,d, both NOR and SOR were positively correlated with RH. This is consistent with high RH favoring the conversion of aerosol precursors gases (e.g., NO_x_ and SO_2_) to NO_3_^−^ and SO_4_^2−^ [1,2,7,9,12]. SOR and NOR were very small (<0.2) when RH was lower than 70% with high O_3_ concentration. In contrast, SOR and NOR were dramatically enhanced to 0.8 and 0.5, respectively, when RH increased from 70% to 100% under low O_3_. Therefore, the high RH was most likely the major reason for the large SOR and NOR, but low O_3_ concentration was not. 

### 3.3. Inter-Comparison within the Whole Observation Period

#### 3.3.1. Difference in Chemical Composition and Conversions

The haze occurred between 19–26 December 2019. The average concentration of OM in the HOP was 22.10 μg m^−3^, which was close to that (23.63 μg m^−3^) during the polluted period between 5–9 December (PP1) and that (21.18 μg m^−3^) during the polluted period between 11–17 December (PP2) before the restriction. In addition, the average total concentrations of nitrogen compounds (NO_x_ + NO_3_^−^: 3.25 µmol m^–3^) and sulfur compounds (SO_2_ + SO_4_^2–^: 0.48 µmol m^–3^) during the HOP were similar to those during PP1 and PP2 (2.75–3.27 µmol m^–3^ and ~0.38 µmol m^–3^). These results indicate that nitrogen and sulfur was weakly influenced by the restriction measures. On the contrary, the average concentrations of NO_3_^−^ and SO_4_^2−^ in the HOP (57.42 and 30.66 µg m^−3^) were higher than those during PP1 (37.07 and 18.02 µg m^–3^) and during PP2 (29.43 and 11.41 µg m^–3^), although the precursor gas levels were lower during the HOP. NO_3_^−^ and SO_4_^2−^ contributed together as much as ~51% of PM_2.5_, which was higher than those in PP1 (~40%) and in PP2 (~39%) before restriction. Therefore, the rapid formations of nitrate and sulfate promoted PM_2.5_ level and caused the severe haze under the restrictions. It was reported that enhanced secondary aerosols might have offset the reduction in primary emissions or decrease in PM_2.5_ in Shanghai, Beijing, and Xi’an [32,43,51,52].

#### 3.3.2. Difference in Source Apportionment and Trajectory Clustering

Coal combustion, industry emission, dust, power, and secondary inorganic aerosols were five PM_2.5_ sources in the PMF analysis (Appendix A). Backward trajectory analysis was then used to investigate the effects of the ranges of source areas influencing the observed PM_2.5_. Three trajectory clusters for each hour in the PP1, PP2, and HOP were acquired via clustering (Figure 5a–c). The proportion of source apportionment of PM_2.5_ corresponding to each trajectory cluster are also presented in Figure 5d–f. The source regions of air mass and source apportionment varied largely in the PP1, PP2, and HOP.

In PP1 before the restriction, 45% of air parcels were from southwestern areas about 300 km away from Zhengzhou (such as Xi’an, Sanmenxia, and Luoyang), where there are few heavy and light industries. The movement of the air mass was slow with average speed 1.7 m s^−1^. About 14% of air parcels came from northern and northwester areas about 2000 km and 1200 km away from the city, where there are few industrial activities. These air masses moved very quickly and passed arid and semi-arid areas. The contribution of mineral dust to PM_2.5_ was large (17%) in comparison to the other two periods. In PP2 before the restriction, 42% of air parcels came from northeastern areas about 500 km away from the city, where there are various heavy and light industries. About 32% of air parcels came from western areas 500 km away from Zhengzhou (such as Xi’an, Yuncheng, and Jiaozuo), which are coal-rich cities. The moving speed of air masses was about 2.9 m s^−1^. The contribution of coal combustion to PM_2.5_ in PP2 was ~34%, indicating that the air mass from those areas was heavily polluted. 

In the HOP, a large proportion of air parcels (52%) moved stagnantly around the southern area about 200 km south at the speed of 1.2 m s^−1^, suggesting that pollutants from the local areas dominated the air pollution (Figure 5c). The PM_2.5_ concentration was approximately four-fold higher than that before the haze formation. Furthermore, both the mass and contribution of coal combustion to PM_2.5_ in this cluster decreased in comparison to PP1 and PP2 (Figure 5), which was due to the restriction. However, the contribution of SIAs (secondary inorganic aerosols) to PM_2.5_ was the highest (~75%) in comparison to those in PP1 and PP2, indicating the formation of SIAs was the major reason for the haze occurrence.

#### 3.3.3. Significance of SIA Formation in HOP

The oxidation pathways for nitrate and sulfate formation were influenced by RH, temperature, and O_3_ [8,46,47,48,49,50]. O_3_ decreased during the HOP and high NOR and SOR were accompanied by low O_3_ (Figure 4). O_3_ was unlikely to have been the major oxidant for the large SIA formation. The RH in the HOP was 78.12%, higher than the 67.86% in PP1 and 51.37% in PP2. The high RH in the HOP was caused by frequent artificial spreading of water vapor in the urban air, a special measure of the Zhengzhou local government to reduce air pollution.

NOR and SOR increased with RH, which is consistent with the fact that high RH favors the conversion of the precursor gases to nitrate and sulfate [1,2,7,9,12]. NOR and SOR were much higher in the HOP than those in PP1 and PP2, when the RH was larger than 70%. (Figure 6). Moreover, the corresponding NO_3_^−^ and SO_4_^2−^ showed obvious rapid increases under RH > 70% than those under RH < 70% (Figure 4). These results indicate high RH (>70%), related to water-vapor spreading during the HOP, which led to the enlargement of the conversion rates of NO_3_^−^ and SO_4_^2−^. Zang et al. (2019) found that sulfate (SO_4_^2−^) and nitrate (NO_3_^−^) were enhanced by approximately 2-fold and 1.5-fold, respectively, under wet conditions [50]. The results indicated that high RH favors the formation of sulfate and nitrate, implying that low RH might help to reduce PM_2.5_ pollution.

If NOR and SOR in the restriction period were not 0.3 and 0.64, but instead were similar to those (0.21 and 0.5) before the restriction (Table 2), the concentration of NO_3_^−^ would have been smaller by 19.04 μg m^−3^ and that of SO_4_^2−^ by 15.32 μg m^−3^. NH_4_^+^ could also have been reduced by 14.65 μg m^−3^. PM_2.5_ levels would have been lower than at least 108.18 μg m^−3^ during the restriction period. These results support the idea that high gas-to-particle conversions in the moisture air under low pollutant emission enhanced NO_3_^−^ and SO_4_^2−^ formations, hindering PM_2.5_ reduction, and consequently leading to the severe haze occurrence [26,51,52].

#### 3.3.4. Inter-Comparisons of the Responses of Secondary Aerosol Formations to Emissions Reductions

A severe haze pollution occurred in Zhengzhou between 19–26 December 2019, though the emergency response measure caused decreases in PM_2.5_ mass from coal combustion by 60–66% and dust by 24–79% compared to those before the restriction. Many studies have reported that the PM_2.5_ level in China decreased by 29.79% from 2016 to 2020, due to reductions on the emissions of NO_x_ and SO_2_, but haze pollution still occurs in northern China in winter [10,20,23,53]. The enhancement of secondary pollution was the major reason for the haze pollution [54,55]. Sulfate and nitrate have changed little in the past decade over the eastern United States, accounting for half of the PM_2.5_ mass, despite a substantial reduction in precursors emissions [56]. SO_4_^2−^ was reduced significantly by 73.3%; however, NO_3_^−^ was reduced relatively less significantly by 29.1%, although emissions of SO_2_ and HNO_3_ in the United States and Canada were significantly reduced by 87.6% and 65.8% from 1990 to 2015 [57].

Results of the present study showed that high SIA concentration formation caused severe haze despite the implementation of emergency response measures to restrict the anthropogenic pollutants. Simulations revealed that limitations of the availability of oxidants relax at lower precursor concentrations, producing particulate matter more efficiently, and weakening the effectiveness of emission reductions over the eastern United States [56]. Due to a notable change in regional chemistry, SOR and NOR increased by more than 50% during the cold season causing the nonlinear relationships between SO_4_^2−^, NO_3_^−^, and their precursors in the United States and Canada [57]. These results suggest that substantial improvements in air quality need larger emission reductions in China, the United States, and Canada.

## 4. Conclusions

In this study, chemical species, positive matrix factorization (PMF), and trajectory clustering were applied to characterize the chemical conversion and explore potential origins of PM_2.5_ to investigate the reasons for the severe haze occurrence in Zhengzhou during 19–26 December, after the emergency response measures were implemented to suppress air pollution in the city.

During the HOP, the average concentration of PM_2.5_ was 172.63 μg m^−3^. The average concentrations of SO_2_ and NO_2_ were 10.23 µg m^−3^ and 66.76 µg m^−3^, which were lower than those in PP1 and PP2. NOR and SOR increased quickly with RH during the HOP when RH was >70%. Large amounts of sulfate (31 µg m^−3^) and nitrate (57 µg m^−3^) were produced under the high RH and contributed to 51% of PM_2.5_ during the HOP. These results indicate that the severe haze was mainly caused by sulfate and nitrate formation via efficient gas-to-particle conversions in moisture air under the restriction of air pollutant emissions. The five PM_2.5_ sources were secondary inorganic aerosols, industry emissions, dust, power, and coal combustion. The contribution of secondary inorganic aerosols to the PM_2.5_ in the HOP was highest (74%). In addition, most air mass came from surrounding areas in the south and moved slowly to the city during the HOP, suggesting the dominance of pollutants from local areas in the PM_2.5_ elevation. In conclusion, high gas-to-particle conversions in the moisture air resulted in efficient formation of secondary inorganic aerosols, and thereby caused the severe haze in Zhengzhou, despite implementation of emergency response measures to restrict the anthropogenic pollutants. The result implies that lowering RH might help to decay and even suppress the formation of SIAs. Therefore, the activities of watering, spraying, and wet sweeping should be limited in order to reduce haze pollution in the city, since these activities must have caused RH growth in the urban air.

## Figures and Tables

**Figure 1 ijerph-19-06405-f001:**
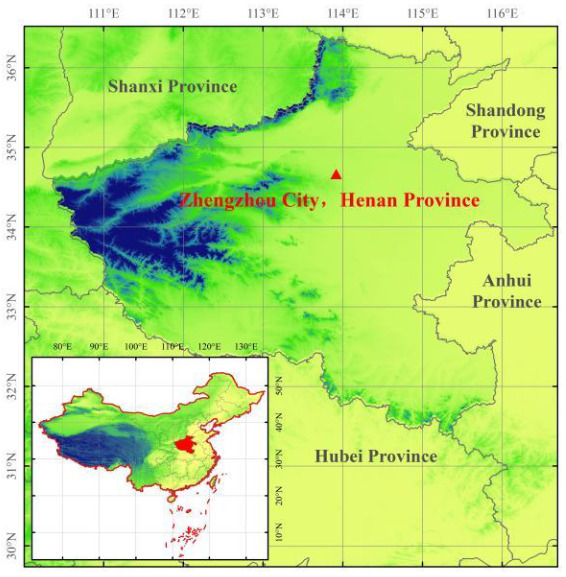
Location of the site (Zhengzhou) and topography in China.

**Figure 2 ijerph-19-06405-f002:**
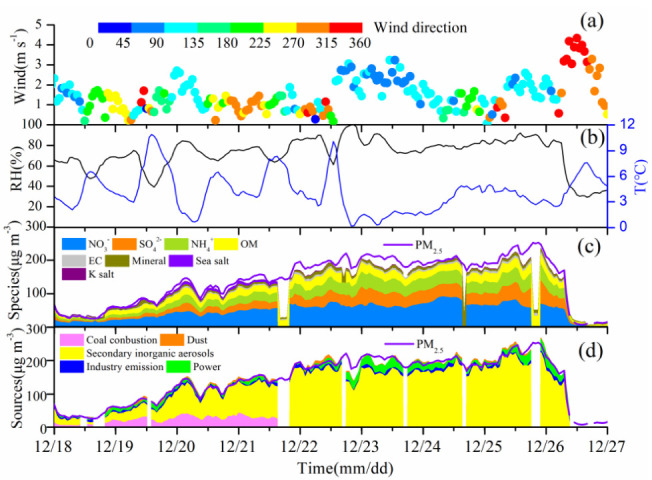
Meteorological conditions (**a**,**b**), concentration and chemical species of PM_2.5_ (**c**), and contributions from each identified source of PM_2.5_ (**d**) during 18−26 December 2019.

**Figure 3 ijerph-19-06405-f003:**
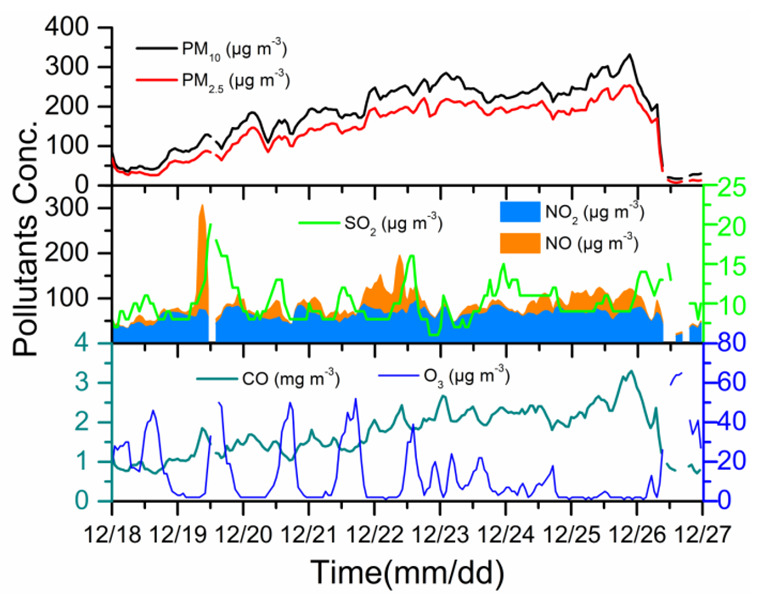
Evolution of particulate matter (PM_10_ and PM_2.5_) and gaseous pollutants (NO, NO_2_, SO_2_, CO, and O_3_) from December 18 to 26.

**Figure 4 ijerph-19-06405-f004:**
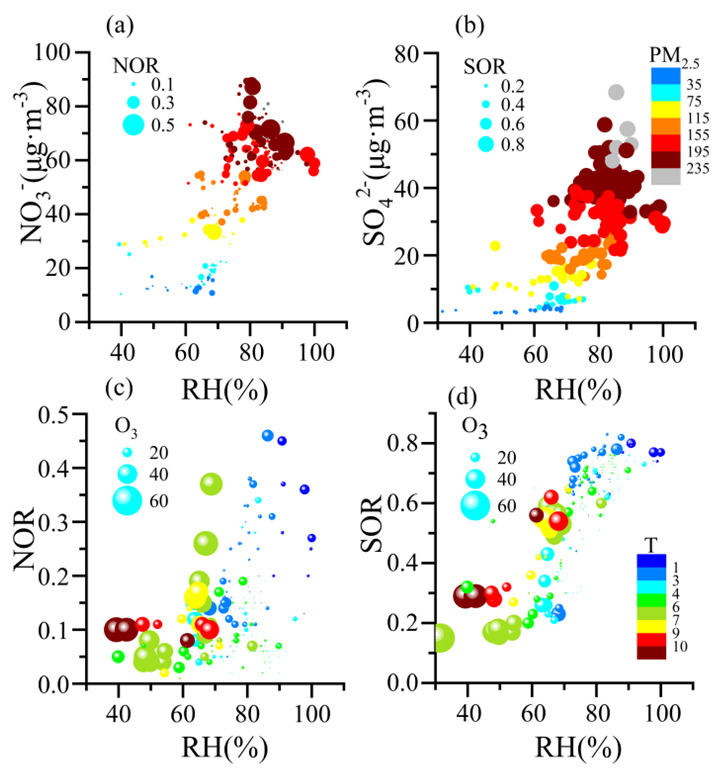
Correlations between RH with NO_3_^−^ (**a**), SO_4_^2−^ (**b**), NOR (**c**), and SOR (**d**). Symbols in (**a**,**b**) are scaled by NOR and SOR and colored by PM_2.5_ concentration; Symbols in (**c**,**d**) are scaled by O_3_ concentration and colored by temperature (T).

**Figure 5 ijerph-19-06405-f005:**
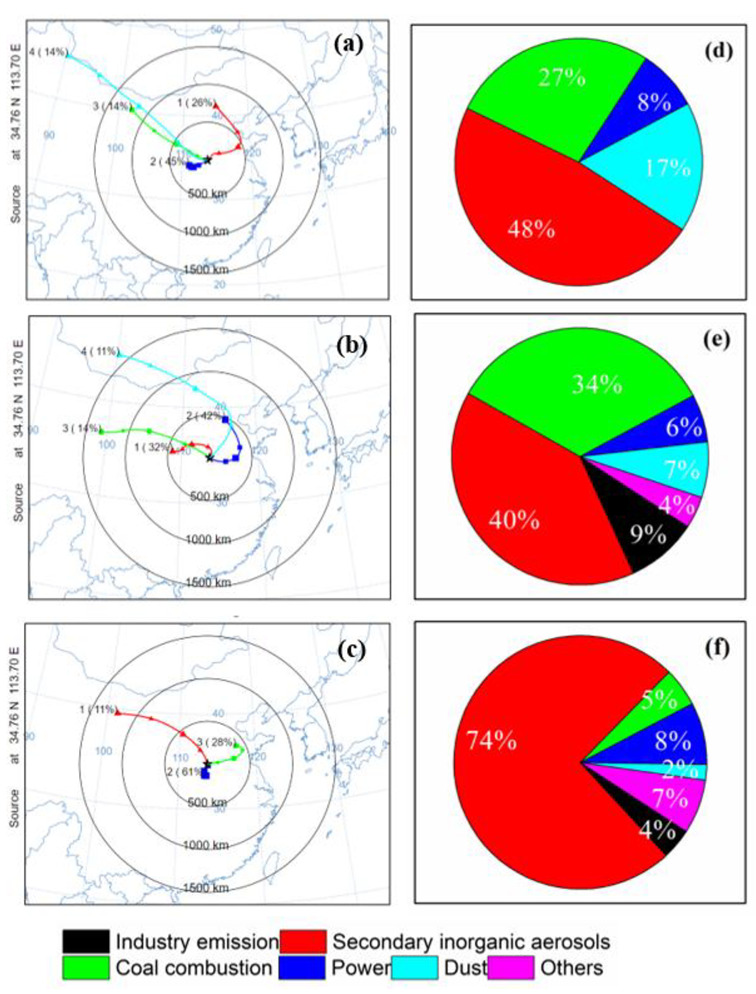
Examples of cluster mean 48 h backward air trajectories and source contribution fractions to PM_2.5_ in PP1 (**a**,**d**), PP2 (**b**,**e**), and HOP (**c**,**f**).

**Figure 6 ijerph-19-06405-f006:**
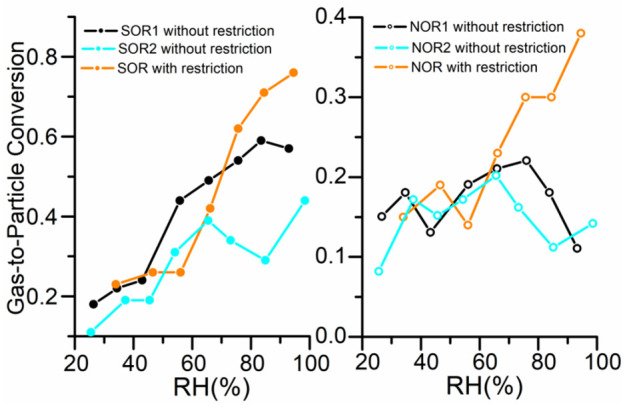
Correlations between SOR, NOR, and RH, and SOR and RH with and without the restriction.

**Table 1 ijerph-19-06405-t001:** Formulas and references of main components in PM_2.5_.

Components or Their Conversions	Formulas	References
Secondary inorganic aerosols (SIAs)	SO_4_^2−^ + NO_3_^−^ + NH_4_^+^	[6,34]
Organic matter (OM)	1.6 × OC	[7,35]
Mineral	2.20 Al + 2.49 Si + 1.63 Ca + 2.42 Fe + 1.94 Ti	[36,37]
Sea salt	Mg^2+^ + Na^+^ + F^−^ + Cl^−^	[38]
K salt	K^+^	[39]
SOR	[SO_4_^2^^−^]/([SO_4_^2^^−^] + [SO_2_]) molar concentration	[40]
NOR	[NO_3_^−^]/([NO_3_^−^] + [NO] + [NO_2_])	[41]

Where [x] is the molar concentration of x species.

**Table 2 ijerph-19-06405-t002:** Average concentration and deviation (in µg m^–3^) of PM_2.5_, PM_2.5_ contents of ions, OM, EC, and precursor gases of SO_2_ and NO_x_; average rates of gas-to-particle conversion (NOR, SOR); and RH (in %) in the three periods of PP1, PP2, and HOP.

	PP1 before Restriction	PP2 before Restriction	HOP with Restriction
PM_2.5_	136.35 ± 37.53	104.162 ± 2.17	172.63 ± 43.10
NO_3_^−^	37.07 ± 10.10	29.43 ± 7.08	57.42 ± 14.98
SO_4_^2−^	18.02 ± 5.39	11.41 ± 4.54	30.66 ± 11.80
NH_4_^+^	20.36 ± 5.68	14.67 ± 3.36	30.88 ± 8.93
OM	23.63 ± 8.43	21.18 ± 4.96	22.10 ± 5.07
EC	3.27 ± 1.29	3.13 ± 1.02	4.83 ± 1.49
NO_x_	122.59 ± 59.43	104.48 ± 44.42	107.12 ± 52.64
SO_2_	12.27 ± 6.29	16.23 ± 5.00	10.23 ± 2.43
O_3_	16.37 ± 22.35	15.81 ± 19.55	9.81 ± 11.62
NOR	0.21 ± 0.08	0.19 ± 0.07	0.30 ± 0.08
SOR	0.50 ± 0.13	0.33 ± 0.14	0.64 ± 0.12
RH	67.86 ± 17.40	51.37 ± 9.99	78.12 ± 10.05

## Data Availability

Not applicable.

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
