# Peer review of "Haze Occurrence Caused by High Gas-to-Particle Conversion in Moisture Air under Low Pollutant Emission in a Megacity of China"

_ijerph, 2022, doi:10.3390/ijerph19116405_

Round 1

Reviewer 1 Report

While the paper is fine in most respects, there have been a number of pieces written on similar topics in the recent past.  As I read, I recalled Zhang et al (2021) in Journal of Environmental Sciences, for example.  The methods similarly have been employed in similar fashion elsewhere.  What is unique about this study - what is its specific contribution that has not been adequately addressed by other work?

Author Response

Response to reviewer 1:Thank you very much. We deeply appreciate your professional comments. We have provided detailed point-by-point responses below in blue font and revised the manuscript accordingly in red font.

Review 1

While the paper is fine in most respects, there have been a number of pieces written on similar topics in the recent past.  As I read, I recalled Zhang et al (2021) in Journal of Environmental Sciences, for example.  The methods similarly have been employed in similar fashion elsewhere.  What is unique about this study - what is its specific contribution that has not been adequately addressed by other work?

Reply: Thank you for this severe comment. We think this misunderstanding is caused by our vague description. Yes, the major conclusion, “the haze was caused by efficient SO2-to-suflate and NOx-to-nitrate conversions under high relative humidity (RH) condition”, sounds not a unique and newly-unwrapped conclusion. However, it occurred when the primary emission of anthropogenic pollutants was obviously reduced at the mega-city Zhengzhou located in the south edge of the North China Plain (NCP). To our knowledge, the occurrence of haze under reduced emission conditions in this area has not been carefully described and discussed. Most published works under reduced emission conditions are on the haze in the north parts of the NCP where the capital city Beijing is located and the eastern part of China where the megacity Shanghai is located. We tried to explore the reasons for the haze occurrence and found the above major conclusion. We raised this conclusion to emphasize that, even under the reduced-emission conditions, haze still occurred due to the efficient conversions, which is obviously different from haze under non-reduced-emission conditions when the major process is secondary formation of inorganic and organic species on pre-existing particles. To make these points clearer, in the revision, we revised the manuscript in Line37-39, Line 81-82, and Line 322-323. 

In addition, if the title of Zhang et al (2021) in the Journal of Environmental Science mentioned by the reviewer was “Characteristics, sources and health risks assessment of VOCs in Zhengzhou, China during haze pollution season”. An investigation of characteristics, sources and health risks assessment of VOCs was carried out at the urban area of Zhengzhou from 1st to 31st December, 2019 in this paper to indicate that vehicular exhaust, industrial processes, combustion, fuel evaporation, and solvent use were five major sources of VOCs, industrial emission was the major contributor to non-carcinogenic, and solvent use was the major contributor to carcinogenic risks. If the title of Zhang et al (2021) in the Journal of Environmental Science mentioned by the reviewer was “Humidity and PM2.5 composition determine atmospheric light extinction in the arid region of northwest China”. The major results in this paper were that relative humidity (RH) played a key role in affecting visibility, and residential coal combustion and vehicle emissions were the major sources of bext. The two papers were different with my paper’ points. We did not find other similar papers in the Journal of Environmental Science mentioned by the reviewer.

Reviewer 2 Report

This paper as much as it contributes to knowledge has made no recommendations based on its findings and this seems not right for a paper that has identified a problem with no solution proposed

Author Response

Response to reviewer 2:Thank you very much. We deeply appreciate your professional comments. We have provided detailed point-by-point responses below in blue font and revised the manuscript accordingly in red font.

Review 2

This paper as much as it contributes to knowledge has made no recommendations based on its findings and this seems not right for a paper that has identified a problem with no solution proposed

(1)Line2-3:This paper as much as it contributes to knowledge has made no recommendations based on its findings and this seems not right

Reply: Thank you for this suggestion. We added recommendations that “Since the artificial water-vapor spreading in the urban air was one of the reasons for the high RH, it is likely that the spreading had unexpected side effects in some certain circumstances and needs to be taken into the consideration in future studies” in Line 37-39 in the revision.

(2)Line19:PM2.5 , Write in full the first time please

Reply: Thank you for the advice. We changed “PM2.5” to “particulate matter with the aerodynamic diameter smaller than 2.5 mm (PM2.5)” in Line 23.

(3)Line32:within past 48 hours

Which 48 hours?

Reply: Thank you for the advice. It was changed to “the last 48 hours movement of the air parcels in December 19 - 26 was stagnant ” in Line 33.

(4)Line36:NCP

write in full the first time please

Reply: Thank you for the advice. NCP was removed in Line 36 and “North China Plain (NCP)” was added in Line 44 in the revision.

(5)Line36:urban air.

so if this happens what are you recommending

Reply: Thank you for the advice. We add some suggests that “Since the artificial water-vapor spreading in the urban air was one of the reasons for the high RH, it is likely that the spreading had unexpected side effects in some certain circumstances and needs to be taken into the consideration in future studies” in Line 37-39.

(6)Line40:PM

What is this PM? it is appearing a second time and it is not clear what it means?

Reply: Thank you for the advice. We changed “PM2.5” to “particulate matter with the aerodynamic diameter smaller than 2.5 mm (PM2.5)” in Line 23.

(7)Line42-43:suppress the heavy air pollution, Chinese government has taken a series of control measures, including cutting down pollutant emissions, limiting car-use, constructing clean-coal power plants, prohibiting open burning of crop residues during harvest sea-

what is the source of this information?

Reply: Thank you for the advice. We cited the studies including the information in Line 48.

(8)Line45-46:As a result, the annual mean PM2.5 concentration significantly reduced nation-wide in the past several years. Source please?

Reply: Thank you for the advice. The data was opened at “http://www.aqistudy.cn/historydata/”. We add some studies in Line 49 in the revision.

(9)Line46-47:Yet, PM2.5 concentration in many megacities of the NCP became high sometimes.

There is a need to add references with empirical evidence here. Saying it is high with no such evidence is not advised in academic writing.

Reply: Thank you for the advice. We have added new references with empirical evidence in Line 50.

(10)Line60-63:However, haze still occurred sometimes in winter. The reduction did not eliminate the occurrence of haze. The nonlinear relationships between particulate matter (PM) and their precursors, unfavorable meteorological conditions, and enhanced secondary production were reported as the causes for the haze [20-22].

There might be a need to merge these sentence to allow for its proper referencing.

Reply: Thank you for the advice. We have merged these sentences and its proper references in Line 63-67.

(11)Line65:areas...

rephrase sentence to read ... and land area of about.

Reply: Thank you for the advice. We changed “areas” to “land area of about” in Line 69.

(12)Line67-70:The city, located in the southern part of the NCP, is surrounded by other densely populated and industrialized cities. Due to the varieties of energy structures and

economy levels from city to city in China, one-city-one air pollution control-policy is urgently required [9].

i think this should be buttressed by a map indicating the exact location of the city to reveal this. You may link it to Fig 1

Reply: Thank you for the advice. We added Fig 1 here.

(13)Line72-74:Zhengzhou implemented emergency response measures, with shutting down and restricting industrial activities, prohibiting open burning, reducing heavy vehicles on road and spreading water vapor on public roads since December 19, 2019 to suppress the occurrence of haze. 

source please?

Reply: Thank you for the advice. The control measures were at http://public.zhengzhou.gov.cn/10K/3525671.jhtml. We cited the link in Line 78.

(14)Line91:measures

measured...correct the tense please

Reply: Thank you for the advice. We did as you suggested.

Reviewer 3 Report

The detailed comments are pasted in the file and maybe consulted while revising the manuscript.
I must say that authors need to put much more energy and time to improve.

Author Response

Response to reviewer 3:Thank you very much. We deeply appreciate your professional comments. We have provided detailed point-by-point responses below in blue font and revised the manuscript accordingly in red font.

Review 3

The detailed comments are pasted in the file and maybe consulted while revising the manuscript.I must say that authors need to put much more energy and time to improve.

(1)Line47:Provide reference please.

Reply: Thank you for this suggestion. New references were added in Line 50.

(2)Line58-64:The authors provide the information only from chinese studies.... It is better to cite some related regional and global studies.

Reply: Thank you for this suggestion. We cited some global studies in Line 62-68.

(3)Line78:what is the novelty of this study... how it is different from other similar case studies?

Reply: Thank you for this suggestion. To our knowledge, the occurrence of haze under reduced emission conditions in this area has not been carefully investigated and discussed. Most published works under reduced emission conditions are on the haze in the north parts of the NCP where the capital city Beijing is located and the eastern part of China where the megacity Shanghai is located. We tried to explore the reasons for the haze occurrence and found the above major conclusion. We raised this conclusion to emphasize that, even under the reduced-emission conditions, haze still occurred due to the efficient conversions, which is obviously different from haze under non-reduced-emission conditions when the major process is secondary formation of inorganic and organic species on pre-existing particles. To make these points clearer, in the revision, we revised the manuscript in Line 37-39, Line 81-82, and Line 322-323. 

(4)Line106-113:Cite some recent studies in which PMF model has been used.

Reply: Thank you for this suggestion. Some recent studies about PMF model were added in Line 116.

(5)Line124-126:fragmented sentence. please rewrite this.

Reply: Thank you for this suggestion. We revised Lines 124-126 into “The wind speed was usually less than 2 m s-1 and the RH was mostly greater than 40% between December 19 and 26, indicating that air movement was stagnant. Under these weather conditions, pollutants were hardly diffused.” in Line 133-135.

(6)Line126-127:How you deduce this statement? please provide some reference.

Reply: Thank you for this suggestion. We revised Lines 124-126 into “ The wind speed was usually less than 2 m s-1 and the RH was mostly greater than 40% between December 19 and 26, indicating that air movement was stagnant. Under these weather conditions, pollutants were hardly diffused.” in Line 133-135.

(7)Line131:Highlight the reasons for this extremely high aerosols load.

Reply: Thank you for this suggestion. We added “The wind speed was1.4 m s-1 and the RH was 89%. SIA concentration was 185 µg m-3, accounting for 73% of PM2.5 concentration. Under the stable weather conditions, high SIA formation was the major reason for the extremely high aerosols load” in Line 138-141.

(8)Line144:why the concentration of SO2 and O3 is high in daytime compared to nighttime. Please provide an argument.

Reply: Thank you for this suggestion. The trend in SOR followed that of RH quite well; it was high during the night but low during the day and more SO2 was converted during nighttime. In comparison, the conversion during daytime was small leading that SO2 concentration was relatively high during the daytime between December 18-22. Because O3 was formed through photochemical reaction, O3 was high in the daytime. These points were added in Line 156-160.

(9)Line146:Please explain why there is an opposite trend of NO2 compared to SO2 & O3 in nighttime/daytime.

Reply: Thank you for this suggestion. NO2 were oxidized by hydroxyl radical (OH) and high O3 to formation HNO3 in daytime, thus NO2 became low in the late afternoon during December 18 - 22. The point was added in Line 163-164.

(10)Line152:Figure 3 has not been cited in the text of this manuscript. Please mention figure 3 in text too.....

Reply: Thank you very much for the careful reading. We made a mistake when editing the draft removing the description with the first appearance of Figure 3. It was added in Line 156 in the revision.

(11)Line153:Increasing rate of what???

Reply: Thank you for this suggestion. “Increasing rate” was changed to “The increases of gas pollutants” in Line 171.

(12)Line166:Cite few studies to justify this narrative.

Reply: Thank you for this suggestion. We have changed this narrative to “ The SO2/CO and NO2/CO ratios decreased with haze pollution development, which was consistent with the substantial increasing PM2.5/CO between December 19- 26 (Figure. S1). This result implies that the air pollution in December 19-26 should have been eliminated although emergency measures had been enacted” in Line 184-186.

(13)Line168:Compare your results with other regional or global studies.

Reply: Thank you for this suggestion. We added “A recent study found that despite emission reductions of 90 % across all sectors over Beijing and surrounding provinces, heavily polluted days with daily mean PM2.5 higher than 225 µg m−3, may not be eliminated to meet the national air quality standards [4]” in Line 186-190.

(14)Line175:There is no figure 1c in this manuscript. Check it and made correction accordingly.

Reply: Thank you for this suggestion. This is an input error. It is Figure 2c. We modified it in the revision.

(15)Line228:compare your results with other studies.

Reply: Thank you for this suggestion. The increasing contribution of sulfate and oxygenated organic aerosol inhibited further PM2.5 reduction during the periods of COVID-19 restriction in Shanghai and Xi’an. It was reported that enhanced secondary aerosol might have offset the reduction in primary emissions or decrease in PM2.5 in Beijing. The descriptions were added in Line 251- 263.

(16)Line284:again compare your results with similar regional studies and provide the reasons for increase/decrease...

Reply: Thank you for this suggestion. Zang et al (2019) found that sulfate (SO42−), and nitrate (NO3) were enhanced by approximately 2-fold, and 1.5-fold respectively under wet conditions. We added these descriptions in the revision in Line 318-322.

(17)Line294:There should be detailed separate discussion section.

Reply: Thank you for this suggestion. Discussions were added in results and discussion section to better support our points. We feel the combination of results and discussion is better than separated discussion section because the targets of each part of discussion are approximately similar, i.e. the major conclusion of this study.

(18)Line315:implemented

Reply: Thank you for this suggestion. “implementation of” was replaced with “implementation of” in Line 354.

Round 2

Reviewer 1 Report

The authors have responded sufficiently to my concerns in the revised version.

Author Response

Thank you very much for your helpful comments.

Reviewer 2 Report

In the conclusion section, there is the need for recommendations and not just a few lines as captured only in the abstract. The essence of your research is not just to identify that there is a problem but to also tell us what you think will and should be done to eliminate the challenges or discoveries you have made 

Author Response

In the conclusion section, there is the need for recommendations and not just a few lines as captured only in the abstract. The essence of your research is not just to identify that there is a problem but to also tell us what you think will and should be done to eliminate the challenges or discoveries you have made

Reply: Thank you for the advice. We added “The result implies that lowering RH might help to decay and even suppress the formation of SIA. Therefore, activities of watering, spray and wet sweeping should be limited in order to reduce haze pollution in the city, since these activities must have caused RH growth in the urban air.” in Line 370-374.

Reviewer 3 Report

I have the following comments and I am sure these shall improve the quality of the manuscript.

The authors provide the information only from Chinese studies.... It is better to cite some related regional and global studies.

what is the novelty of this study... how is it different from other similar case studies?

There should be a detailed separate discussion section.

Author Response

I have the following comments and I am sure these shall improve the quality of the manuscript.

The authors provide the information only from Chinese studies.... It is better to cite some related regional and global studies.

what is the novelty of this study... how is it different from other similar case studies?

There should be a detailed separate discussion section

Reply: Thank you for this suggestion again. We added a detailed separate discussion section, including some related regional and global studies in Line 325-348. The detailed separate discussion section was below:

“3.3.4 Inter-comparisons of the responses of secondary aerosol formations to emissions reductions

Haze formed occurred in Zhengzhou between December 19-26, 2019, although the emergency response measure caused decreases in PM2.5 mass from coal combustion by 60%-66% and dust by 24%-79%, compared with those before the restriction. Many studies have reported that PM2.5 level in China decreased by 29.79% from 2016 to 2020, due to reductions in the emissions of NOx and SO2, but haze pollution till occurs in northern China in winter [10,20,23,53]. The enhancement of secondary pollution was the major reason for the haze pollution [54-55].

Sulfate and nitrate have changed little in the past decade over the eastern United States, accounting for half of the PM2.5 mass, despite a substantial reduction in precursors emissions [56]. SO42− was reduced significantly by 73.3 %, however, NO3 was reduced relatively less significant by 29.1 %, although emissions of SO2 and HNO3 in the United States and Canada were significantly reduced by 87.6 % and 65.8 % from 1990 to 2015 [57].

Results of the present study showed that high SIA concentration formation caused severe haze despite the implementation of the emergency response measures to restrict the anthropogenic pollutants. Simulations revealed that limitations of the availability of oxidants relax at lower precursor concentrations, producing particulate matter more efficiently, and weaken the effectiveness of emission reductions over the eastern United [56]. Due to a notable change in regional chemistry, the SOR, and NOR increased by more than 50% during the cold season caused the nonlinear relationships between SO42−, NO3, and their precursors in the United States and Canada [57]. These results suggest that the substantial improvements air quality need larger emission reductions in China, United States, and Canada.”